# Zinc Oxide–Incorporated Chitosan–Poly(methacrylic Acid) Polyelectrolyte Complex as a Wound Healing Material

**DOI:** 10.3390/jfb14040228

**Published:** 2023-04-17

**Authors:** David Sathya Seeli, Abinash Das, Mani Prabaharan

**Affiliations:** Department of Chemistry, Hindustan Institute of Technology and Science, Padur, Chennai 603 103, India

**Keywords:** polyelectrolyte complexes, chitosan, poly(methacrylic acid), wound healing, zinc oxide

## Abstract

A novel type of porous films based on the ZnO-incorporated chitosan–poly(methacrylic acid) polyelectrolyte complex was developed as a wound healing material. The structure of porous films was established by Fourier-transform infrared spectroscopy (FTIR), X-ray diffraction (XRD), and energy dispersive X-ray (EDX) analysis. Scanning electron microscope (SEM) and porosity studies revealed that the pore size and porosity of the developed films increased with the increase in zinc oxide (ZnO) concentration. The porous films with maximum ZnO content exhibited improved water swelling degree (1400%), controlled biodegradation (12%) for 28 days, a porosity of 64%, and a tensile strength of 0.47 MPa. Moreover, these films presented antibacterial activity toward *Staphylococcus aureus* and *Micrococcus* sp. due to the existence of ZnO particles. Cytotoxicity studies demonstrated that the developed films had no cytotoxicity against the mouse mesenchymal stem (C3H10T1/2) cell line. These results reveal that ZnO-incorporated chitosan-poly(methacrylic acid) films could be used as an ideal material for wound healing application.

## 1. Introduction

Dermal damages are increasing globally due to accidents, diabetic ulcers, surgical procedures, intravenous ulcers, and blisters. The concern of dermal wounds imparts a nonhealing sore and a serious health issue to the human body. In order to avoid massive loss of blood, injured skin due to disease or accidents has to be secured by wound healing materials, thereby enhancing wound restoration. Additionally, chronic wounds, which are susceptible to long-lasting bacterial infection, trauma, cancerous ulcers, and skin diseases, need efficient wound dressings [1,2]. Among the currently available wound dressing materials, most of them are less effective because of insufficient mechanical strength, low cell penetration behavior, high threat of infection, and lower shelf life. In this regard, the choice of an ultimate wound bandage material that is biodegradable, biocompatible, flexible, mechanically strong, adhesive, and appropriate for use is the need of the hour. The ideal wound dressing material should retain moisture, allow gaseous exchange, and safeguard the wounded area against microbial contagions [3,4,5]. As wound healing processes, such as oxidative destruction of pathogens, re-epithelialization, angiogenesis, and synthesis of collagen, need oxygen, it is necessary to maintain adequate oxygen permeability in the wound site.

Over the past few years, polyelectrolyte complexes (PECs) developed by mixing oppositely charged polymers have received great attention as wound dressing materials because of their exceptional physicochemical and biological characteristics [6]. The formation of PECs by the interaction of polyanion and polycation through liquid–liquid phase separation in colloidal solution takes place spontaneously either in bulk or at the interface. The stability and morphology of formed PECs are influenced by a number of extrinsic and intrinsic elements, such as charge density, polyelectrolyte concentration, chain flexibility, and molar mass. In addition to the above factors, parameters such as temperature, pH, and ionic strength also play a major role in the formation of PECs [7]. Over the past few decades, chitosan has been considered to be promising material for wound healing due to its biocompatibility, biodegradability, nontoxic, and nonimmunogenic properties. Chitosan, a natural-based biopolymer, consists of *D*-glucosamine and *N*-acetyl-*D*-glucosamine units bonded by (1–4) glyosidic linkage attained by the *N*-deacetylation of chitin. The primary amino group present in chitosan is responsible for the formation of PECs in various forms, such as membranes, hydrogels, beads, nanoparticles, and microparticles. The protonated amino group of chitosan interacts with carboxyl groups of anionic polymers, such as carboxymethyl cellulose, alginate, collagen, xanthan gum, and pectin, through various forces of attraction, such as electrostatic interaction, van der Waals force, hydrophobic interaction, and hydrogen bonding, leading to the formation of PECs [8]. Currently, poly(methacrylic acid) (PMA) has been widely considered for biomedical applications due to its nontoxicity and biocompatibility. Due to the presence of abundant carboxyl groups, PMA can be chemically modified or conjugated with other biomaterials to extend its functionality as bioactive material [9]. The antibacterial property, which is essential for effective wound healing, can be obtained by incorporating metal oxide nanoparticles in the matrix of PECs. Among the metal oxides, nano ZnO possesses excellent antibacterial and antifungal activity. Hence, nano ZnO was incorporated into wound dressing materials to enhance their antimicrobial property. Since the leaching of Zn^2+^ ions from nano ZnO would help in keratinocyte migration towards the wound site and promote epithelialization, it has a wide scope to develop potential wound dressing materials based on polymer composites [10].

The aim of this work is to prepare three-dimensional (3D) porous wound healing films with increased mechanical strength and antibacterial properties based on chitosan–PMA PECs incorporated with various amounts of ZnO nanoparticles (CPMAZnO) by the lyophilization method. In this study, the developed CPMAZnO films were scrutinized by FTIR, X-ray diffraction, SEM, and EDX analysis. The suitability of CPMAZnO films as potential wound healing material was evaluated by measuring its porosity, swelling degree, biodegradability, biocompatibility, antimicrobial properties, and live–dead cell assay.

## 2. Materials and Methods

### 2.1. Materials

Chitosan (MW, ~250 kDa, and degree of deacetylation, ~75%), methacrylic acid, potassium persulfate (K_2_S_2_O_8_), glacial acetic acid, zinc acetate dihydrate, and sodium hydroxide were bought from Sigma-Aldrich, Mumbai, India. The C3H10T1/2 cell line was acquired from the National Centre for Cell Science, Pune, India. All other reagents utilized were of analytical grade.

### 2.2. Synthesis

#### 2.2.1. Preparation of PMA

PMA was synthesized by radical homopolymerization using 10 g of methacrylic acid in 20 mL of distilled water and 0.1 mg of K_2_S_2_O_8_ as an initiator. During the reaction, the initiator was added slowly to the 20 mL of monomer solution under uniform stirring and a nitrogen environment at 70 °C for 24 h. Thereafter, the resultant PMA was purified by the dialysis method using distilled water and a cellulose acetate membrane (MWCO, 10 kDa). Finally, the product was separated from the solution by the freeze-drying method and stored in a desiccator for further use.

#### 2.2.2. Preparation of CPMA and CPMAZnO Films

Initially, CPMA PEC was prepared by treating 2 wt. % chitosan dissolved in 100 mL of 1% glacial acetic acid with 100 mL of 2 wt. % aqueous PMA solution under constant stirring. After 24 h, the formed CPMA PEC was neutralized using 0.2 M NaOH solution and then transferred into a petri dish, followed by freeze-drying to obtain CPMA films. To prepare CPMAZnO films, initially, CPMA PEC was prepared and mixed with varying concentrations of a zinc acetate dihydrate precursor (50, 75, 100 and, 150 mmol) under stirring at 70 °C for 3 h. Thereafter, the resultant mixture was neutralized with 0.2 M NaOH solution and then poured into a petri dish (35 mm × 10 mm), followed by freeze-drying at −50 °C and 0.01 mbar pressure to obtain Zn(OH)_2_-loaded CPMA films. Finally, the formed films were dried at 120 °C for 3 h to produce CPMAZnO films. Table 1 shows the amounts of reagents used for the formation of various types of CPMAZnO films.

#### 2.2.3. Characterization Methods

The chemical composition of the prepared materials was established by FTIR spectral analysis. The FTIR spectra were recorded on a double-beam PerkinElmer 1600 FTIR spectrometer. The microstructure of the products was examined with a wide-angle X-ray diffractometer (Brucker’s D-8) using Ni-filtered Cu Ka radiation as an X-ray source. The morphology and elemental composition of the films were investigated with a SEM (COXEM TM200) at 15 kV. The porosity of the films was evaluated by the fluid displacement method [11]. The tensile strength and elongation at break of the films with dimensions of 4 × 1 × 0.3 cm^3^ were measured by the ASTM D882–02 standard.

### 2.3. Swelling Studies

The swelling characteristic of the prepared materials was studied in phosphate-buffered saline (PBS) at pH 7.4 and 37 °C. The precisely weighed (*W*_0_) samples were submerged in 50 mL of PBS. At fixed time periods (0.5, 2, 3, 5, 8, 12, 18, and 24 h), the weights (*W_t_*) of swollen samples were determined after the removal of surplus water. The swelling degree of the PEC films at time *t* was determined using the following formula [12]:Swelling degree (%) = (*W_t_* − *W*_0_)/*W*_0_ × 100 
where *W_t_* and *W*_0_ are the weight of a wet sample at time *t* and the weight of a dry sample, respectively.

### 2.4. In Vitro Biodegradation Studies

The known weights of prepared films were engrossed in 50 mL of PBS solution consisting of 25 mg of lysozyme at 37 °C for 21 days. After 7, 14, and 21 days, each set of films was removed from the medium and washed with deionized water, followed by freeze-drying. The degradation percentage of the films was assessed using the following formula [13]:Degradation (%) = (*W_i_* − *W_t_*)/*W_i_* × 100
where *W_i_* and *W_t_* are the dry weights of the samples before and after degradation, respectively.

### 2.5. Antibacterial Studies

Wound infection on the skin surface is most often caused by bacteria, such as *Staphylococcus aureus* (*S. aureus*) and *Micrococcus* sp. Hence, in this study, the antibacterial property of CPMAZnO films against *S. aureus* and *Micrococcus* sp. was determined using the diffusion method. About 0.1% inoculum deferments of test bacterial cultures were spread evenly over the surface of an agar plate. Then, 6 mm wells were cut on the agar surface of each plate under sterile circumstances. Thereafter, known weights of CPMAZnO films were placed on the plates and stored at 37 °C for 48 h. Finally, the zone of inhibition across the wells of each film was determined.

### 2.6. MTT and Live–Dead Cell Assay

The biocompatibility of CPMAZnO films was evaluated using C3H10T1/2 cells by indirect MTT assay [14]. Initially, 10 mg of the films were incubated with 1 mL of DMEM containing 10% FBS for 24 h. Thereafter, the conditioned medium was removed from the wells and analyzed its biocompatibility against the C3H10T1/2 cells. Briefly, 1 × 10^4^ cells were cultured in each well of 96-well plates and incubated for 24 h. Followed by incubation, the medium was replaced with 100 μL of the conditioned medium and further incubated for 24 h. Thereafter, 10 μL of MTT (5 mg/mL in PBS) was added to the wells and incubated in the dark for 3 h to produce formazan crystals. Then, 100 μL of DMSO was added to each well to dissolve the formed formazan crystals. The absorbance of the resulting solutions was determined at 570 nm using a Tecan Infinite M200 PRO microplate reader, Tecan, Männedorf, Switzerland. All the experiments were conducted in triplicate. Statistical analysis was performed using SPSS 26 software by one-way ANOVA and the Student–Newman–Keuls test. A *p*-value ˂ 0.01 was considered statistically significant. Live–dead cell assay was performed using the dual acridine orange/ethidium bromide (AO–EB) staining test [15]. The C3H10T1/2 cells (5 × 10^3^ cells/well) were cultured in 96-well plates at 37 °C in 5% CO_2_ and a 95% air humidified environment for 24 h. Thereafter, the cells were mixed with 10 mL of PBS containing 100 mg of CPMA and CPMAZnO films, respectively, for 48 h. Thereafter, the medium was separated, and the AO–EB staining agent (25 μL) was added with the cells. The fluorescence intensity of the cells was detected by a fluorescence microscope.

## 3. Results and Discussion

### 3.1. Preparation and Characterization of CPMAZnO Films

The CPMAZnO films were developed using a four-step process. Initially, PMA was synthesized by free radical polymerization of methacrylic acid under a nitrogen atmosphere using potassium persulfate as an initiator for 24 h at 70 °C. Second, the PMA and chitosan solutions were blended together at a ratio of 2:2 wt. % under constant stirring for 24 h. As the amino groups of chitosan were protonated into NH_3_^+^ groups at pH 5–6, the negatively charged carboxyl (COO^-^) groups of PMA formed PEC with chitosan due to the strong electrostatic interactions [16]. In the third step, chitosan/PMA/zinc acetate films were prepared from chitosan/PMA PEC containing different amounts of zinc acetate dihydrate. Then, the obtained chitosan/PMA/zinc acetate films were treated with 0.2 M NaOH solution for complete neutralization and, consequently, freeze-dried after cleaning with deionized water to form Zn(OH)_2_-loaded chitosan/PMA films. Finally, the Zn(OH)_2_-loaded chitosan/PMA films were dried at 120 °C for 3 h to form CPMAZnO porous films [17]. Figure 1 illustrates a schematic representation for the preparation of CPMAZnO films.

The chemical composition of CPMAZnO films was assessed and related with that of pure chitosan, PMA, and CPMA films by FTIR analysis. Figure 2 shows the FTIR spectrum of chitosan, PMA, CPMA, and CPMAZnO films. In the spectrum of chitosan (A) in Figure 2, a distinctive broad band at 3452 cm^−1^ was observed due to the hydroxyl groups present in chitosan. The bands noted at 2920 and 2860 cm^−1^ could be attributed to the existence of aliphatic groups. The characteristic absorption peaks at 1662, 1593, and 1366 cm^−1^ corresponded to C=O stretching (amide I), N−H bending (amide II), and C−N stretching (amide III), respectively [18]. In the spectrum of PMA (B) in Figure 2, the characteristic absorption bands appeared at 3485, 2930, and 2890 cm^−1^ due to –OH stretching and aliphatic CH stretching vibrations [19]. The absorption peaks at 1490, 1455, and 1389 cm^−1^ represent the C-OH bending vibration, methyl, and methylene group of PMA, respectively. The strong peaks at 1641 and 1152 cm^−1^ were attributed to the stretching vibration of carbonyl groups present in PMA [20]. In the FTIR spectrum of the CPMA film (C) in Figure 2, the characteristic bands of chitosan and PMA were observed. The broad absorption band of chitosan due to -OH and -NH stretching has become narrow, and a new band shift has been noted at 1403 and 1110 cm^−1^. These observations showed that the NH_3_^+^ group of chitosan has been involved in bonding with the COO^−^ group of PMA in the CPMA film [21]. The spectrum of the CPMAZnO film (D) in Figure 2 showed the bands of chitosan and PMA with a slight movement of the carboxylate peak to 1638 cm^−1^. The broad band noted at 3400 cm^−1^ was due to the -OH stretching vibrations. It is noted that the -OH band narrowed in the CPMAZnO film due to the development of intermolecular hydrogen bonding and the electrostatic attraction among the two polymer chains. Moreover, the presence of an absorption band at 619 and 1409 cm^−1^ indicated the successful incorporation of ZnO particles within the CPMA film [9].

The crystalline structure of CPMAZnO films was analyzed and correlated with that of chitosan, PMA, and CPMA films using the XRD technique. In the spectrum of chitosan (A) in Figure 3, two typical diffraction peaks corresponding to 8° and 21° were observed due to the semicrystalline nature of chitosan. A major peak noted at 21° could be attributed to form II of the rhombic crystal of chitosan [22]. In the XRD pattern of PMA (B) in Figure 3, the characteristic halo peaks were observed at 16° and 32°, which indicated the amorphous nature of PMA [23]. The spectrum of the CPMA film (C) in Figure 3 showed a broad halo peak in between 20° and 40°, which indicated that the semicrystalline nature of chitosan was diminished due to the formation of amorphous PEC with PMA. The XRD pattern of CPMAZnO films (D-G) in Figure 3 demonstrated the in situ formation of ZnO within the CPMA matrix when increasing the concentration of a zinc acetate dihydrate precursor from 50 to 150 mmol in the polymer mixture. In the spectrum of the CPMAZnO-4 film, the peaks observed at 33.6°, 56.7°, and 59.4° could be attributed to (002), (110), and (103) crystal planes of ZnO, respectively [24,25]. From the XRD spectrum of the CPMAZnO-4 film, the crystallite size of ZnO was determined as 11 nm using the Debye–Scherrer formula.

Figure 4 illustrates the EDX spectrum of control CPMA and CPMAZnO films. The EDX spectrum of the CPMA film showed the peaks related to C, O, and N because of the presence of chitosan and PMA (Figure 4A). In the EDX spectra of CPMAZnO films, the bands due to Zn atoms were observed along with the bands of C and O atoms (Figure 4B–E). Moreover, the intensity of the bands due to Zn atoms was found to be higher in CPMAZnO-4 films than that of other CPMAZnO films, which indicated the existence of a higher amount of ZnO in the CPMAZnO-4 polymer matrix. As the relative percentage of N atoms in CPMAZnO films intensely reduced due to the increase in the proportion of ZnO, the band due to N atoms was not predominantly observed in the spectra of CPMAZnO films [26]. The amount of Zn atom in the CPMAZnO-1, CPMAZnO-2, CPMAZnO-3, and CPMAZnO-4 films was 9.54, 12.54, 15.56, and 22.64 wt. %, respectively. This observation once again confirms the presence of ZnO in the CPMAZnO films.

### 3.2. Morphology and Porosity of CPMAZnO Films

Figure 5 displays the SEM pictures of CPMA and CPMAZnO films. The CPMA films presented rough morphology without any pores on the surface (Figure 5A). This observation can be due to the rise in polymer concentration, which determined a major challenge to pore growth on the sample surface. Meanwhile, in the case of CPMAZnO films (Figure 5B–E), the micropores appeared on their surface, and the size of the pores increased with the increase in ZnO content. The CPMAZnO-4 films showed the well-interconnected pores with a mean size of 40 µm, which could be appropriate for the cell attachment, growth, and proliferation. The increased pore size of CPMAZnO-4 films may be due to the presence of hydrophilic ZnO. ZnO could provide sites for water absorption, which results in the formation of larger ice crystals within the polymer matrix during the freezing. Later, these larger ice crystals tend to form the pores with a larger size upon drying.

Porosity is one of the preferred properties of wound healing materials for an effective wound healing process. The optimal level of porosity that requires the diffusion of oxygen and nutrients would be desirable for the cell binding, growth, and proliferation of wound healing materials. Hence, in this study, the porosity of CPMAZnO films was determined to decide their suitability as wound healing materials. The porosity measurement based on the alcohol displacement method revealed that as the concentration of ZnO increases, the porosity of the CPMAZnO film increases from 8 ± 2 to 64 ± 3%. Appropriately, the porosity of CPMA, CPMAZnO-1, CPMAZnO-2, CPMAZnO-3, and CPMAZnO-4 film was 8 ± 2, 24 ± 4, 35 ± 3, 58 ± 4, and 64 ± 3%, respectively. This observation may be due to the strong interaction of ZnO with the COO^-^ ions of PMA. It was reported that the best porosity for wound healing material is in the range of 60–90% [27]. Since the CPMAZnO-4 film exhibited adequate porosity (64 ± 3%), that could be more suitable for effective wound regeneration.

### 3.3. Swelling Behavior of CPMAZnO Films

Figure 6 shows the swelling behavior of CPMA and CPMAZnO films in PBS at 37 °C for 24 h. In general, the swelling behavior of both CPMA and CPMAZnO films improved with the increase in time due to the presence of hydrophilic groups in chitosan and PMA polymer chains. However, the swelling behavior of CPMAZnO films was improved when compared with that of CPMA films and influenced by the ZnO concentration. The CPMA film presented rapid swelling initially and reached equilibrium after 8 h with a maximum swelling degree of 770%. Meanwhile, in the case of CPMAZnO films, the swelling degree improved as an increase in ZnO content. Accordingly, the CPMAZnO-1, CPMAZnO-2, CPMAZnO-3, and CPMAZnO-4 films exhibited a maximum swelling degree of 870%, 1210%, 1350%, and 1400%, respectively. The presence of hydrophilic groups along with the development of micropores when increasing the ZnO content in the films might be the reason for this observation. This result indicates that the established CPMAZnO films could be apt for wound regeneration because of their higher water swelling and retention ability.

### 3.4. Tensile Strength and Biodegradation Studies

The perfect wound healing materials should have adequate tensile strength for their practical use in wound regeneration process. Hence, the tensile strength and elongation at break of CPMA and CPMAZnO films were determined to evaluate their mechanical properties. The CPMA film presented a tensile strength and elongation at break of 0.23 ± 0.13 MPa and 24 ± 3.4%, respectively. However, when increasing the ZnO concentration, the tensile strength and elongation at break of the samples increased noticeably. Accordingly, the tensile strength and elongation at break of CPMAZnO-1, CPMAZnO-2, CPMAZnO-3, and CPMAZnO-4 films were 0.28 ± 0.15, 0.32 ± 0.11, 0.39 ± 0.15, and 0.47 ± 0.12 MPa and 32 ± 2.5, 37 ± 3.2, 44 ± 2.9, and 53 ± 2.4%, respectively. The strong interaction between the polymer chains by ZnO and thereby the increase in polymer stiffness could be the reason for these observations.

The controlled biodegradation behavior is highly desirable to the wound healing materials for effective tissue regeneration and their ease of removal from the wound site. Hence, the biodegradation profile of CPMAZnO films was assessed and related with that of CPMA film in 50 mL of PBS containing lysozyme (25 mg) at 37 °C for 30 days (Figure 7). It was found that the control CPMA films completely (100%) disintegrated in the medium, while CPMAZnO-1, CPMAZnO-2, CPMAZnO-3, and CPMAZnO-4 films degraded only by about 80%, 55%, 47%, and 12%, respectively, after 30 days of treatment. This observation demonstrates that the controlled biodegradation behavior of CPMAZnO films was increased with the increase in ZnO content because of the interaction among the polymer chains by ZnO ions.

### 3.5. Antibacterial and Cytotoxicity Studies

Figure 8 depicts the antibacterial activity of CPMA and CPMAZnO films. The control CPMA film showed no inhibition zone against *S. aureus* and *Micrococcus* sp. bacteria due to its neutral nature. Among the CPMAZnO films, the CPMAZnO-4 film exhibited antibacterial activity against both test bacteria at a significant level. The inhibitory zone developed by the CPMAZnO-4 film was 16 and 18 mm for *S. aureus* and *Micrococcus* sp., respectively. The CPMAZnO-4 film showed higher antibacterial activity towards *Micrococcus* sp. as compared with *S. aureus*, which was due to the thicker peptidoglycan layer of *S. aureus*. A similar result was observed in the flexible and microporous ZnO-incorporated chitosan composite bandages [28]. The antibacterial activity of the CPMAZnO-4 film indicated that this film could be appropriate for effective wound healing.

Figure 9 displays the C3H10T1/2 cell viability of control TCPS, CPMA, and CPMAZnO films for 24 h. As shown, the CPMA film did not exhibit any adverse effects on the cell viability for up to 24 h when compared with the control TCPS. However, when increasing the content of ZnO, the CPMAZnO films showed a marginal decrease in cell viability, which may be due to the toxicity of ZnO by endoplasmic reticulum stress, cytotoxicity, and genotoxicity associated with reactive oxygen species generation [29]. Accordingly, the cell viability of the CPMAZnO-1, CPMAZnO-2, CPMAZnO-3, and CPMAZnO-4 films was found to be 99%, 99%, 98%, and 96%, respectively. The insignificant decrease in cell viability of CPMAZnO films compared with that of control films indicated that the developed CPMAZnO films are nontoxic to the cultured C3H10T1/2 cell line. This finding is consistent with that of chitosan/pectin/ZnO against the primary human dermal fibroblast cells [26].

The cell viability in the presence of CPMAZnO was further assessed by live–dead cell assay using the AO–EB dual staining test (Figure 10). The C3H10T1/2 cells treated with all types of CPMAZnO films exhibited the well-defined spherical shape with uniform green fluorescence due to their live state as observed for the control CPMA film. These results demonstrate that the CPMAZnO films could be safe and, hence, potentially serve as a wound healing material.

## 4. Conclusions

Novel types of porous wound healing films based on chitosan, PMA, and ZnO were prepared and characterized by FTIR, XRD, SEM, and EDX. The porosity, swelling, and biodegradation behaviors of the prepared CPMAZnO films were found to be influenced by the concentration of ZnO. Because of the existence of a higher amount of ZnO, the CPMAZnO-4 films exhibited adequate porosity (64 ± 3%) and interconnected pores with an average size of 40 µm, which could be appropriate for enhanced cell growth and proliferation. Among the developed films, the CPMAZnO-4 film presented a higher swelling degree (1400%) in 24 h and controlled biodegradation (~12%) for 30 days due to its higher porosity and crosslinking density, respectively. The antibacterial study revealed that the CPMAZnO-4 film had antibacterial activity against *S. aureus* and *Micrococcus* sp. with an inhibition zone of 16 and 18 mm, respectively. The MTT assay demonstrated that the developed CPMAZnO films are nontoxic to C3H10T1/2 cells irrespective of ZnO content. These results indicate that the CPMAZnO films could be safe and potentially used as effective wound healing materials. Incorporating anti-inflammatory drugs into the developed CPMAZnO films and evaluating their drug release and drug permeation abilities through the skin will be the future strategy to assess the possibility of expending these films as versatile wound dressing materials. The combination of drugs into CPMAZnO films will expand the application scope of the dressing materials, enabling targeted treatments of skin contagions, deep wounds, and burns, while the 3D network of the films can enhance the bioavailability of drugs by controlling their release profile.

## Figures and Tables

**Figure 1 jfb-14-00228-f001:**
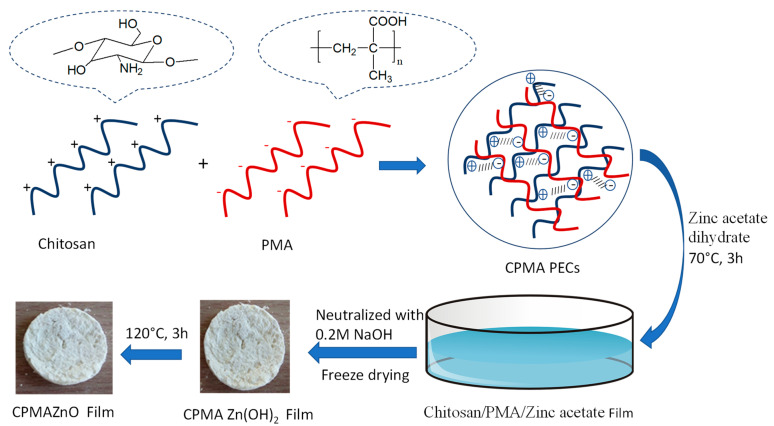
Reaction scheme for the preparation of CPMAZnO PEC films.

**Figure 2 jfb-14-00228-f002:**
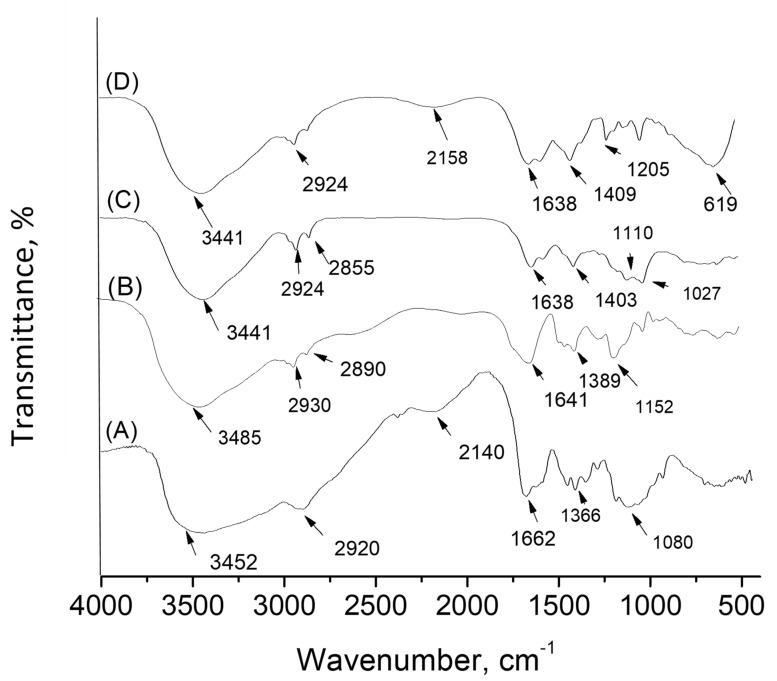
FTIR spectrum of (A) chitosan, (B) PMA, (C) CPMA, and (D) CPMAZnO films.

**Figure 3 jfb-14-00228-f003:**
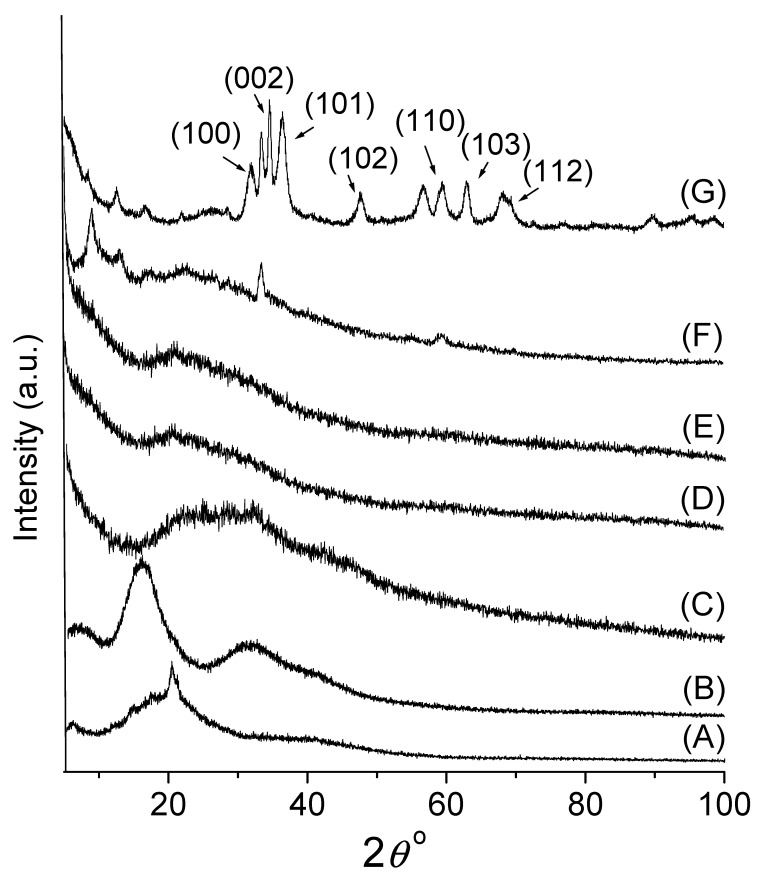
XRD pattern of (A) chitosan, (B) PMA, (C) CPMA, (D) CPMAZnO-1, (E) CPMAZnO-2, (F) CPMAZnO-3, and (G) CPMAZnO-4 films.

**Figure 4 jfb-14-00228-f004:**
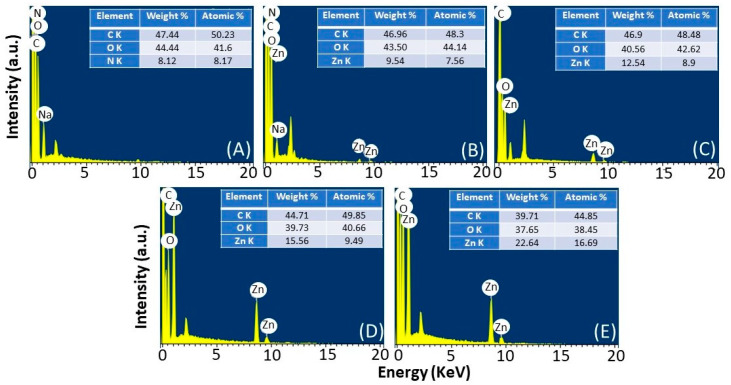
EDX image of (**A**) CPMA, (**B**) CPMAZnO-1, (**C**) CPMAZnO-2, (**D**) CPMAZnO-3, and (**E**) CPMAZnO-4 films.

**Figure 5 jfb-14-00228-f005:**
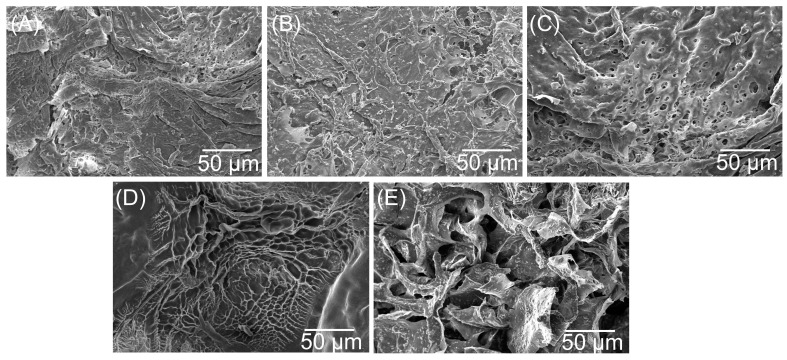
SEM picture of (**A**) CPMA, (**B**) CPMAZnO-1, (**C**) CPMAZnO-2, (**D**) CPMAZnO-3, and (**E**) CPMAZnO-4 films.

**Figure 6 jfb-14-00228-f006:**
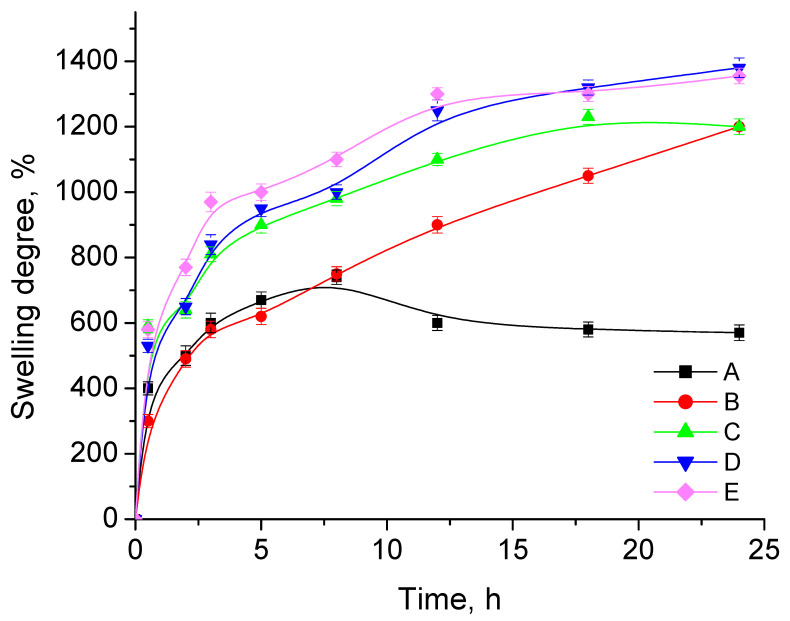
Swelling degrees of (A) CPMA, (B) CPMAZnO-1, (C) CPMAZnO-2, (D) CPMAZnO-3, and (E) CPMAZnO-4 films.

**Figure 7 jfb-14-00228-f007:**
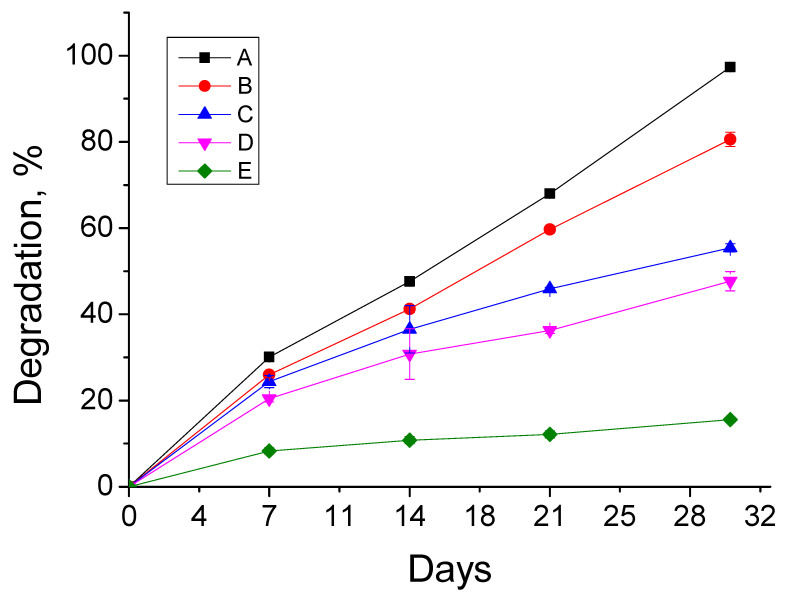
Biodegradation behavior of (A) CPMA, (B) CPMAZnO-1, (C) CPMAZnO-2, (D) CPMAZnO-3, and (E) CPMAZnO-4 films.

**Figure 8 jfb-14-00228-f008:**
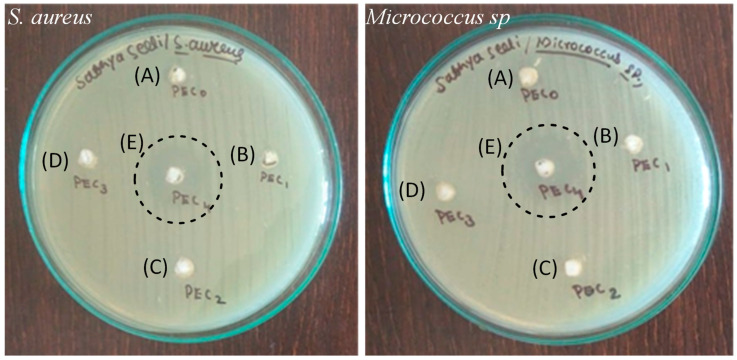
Antibacterial activity of (A) CPMA, (B) CPMAZnO-1, (C) CPMAZnO-2, (D) CPMAZnO-3, and (E) CPMAZnO-4 films.

**Figure 9 jfb-14-00228-f009:**
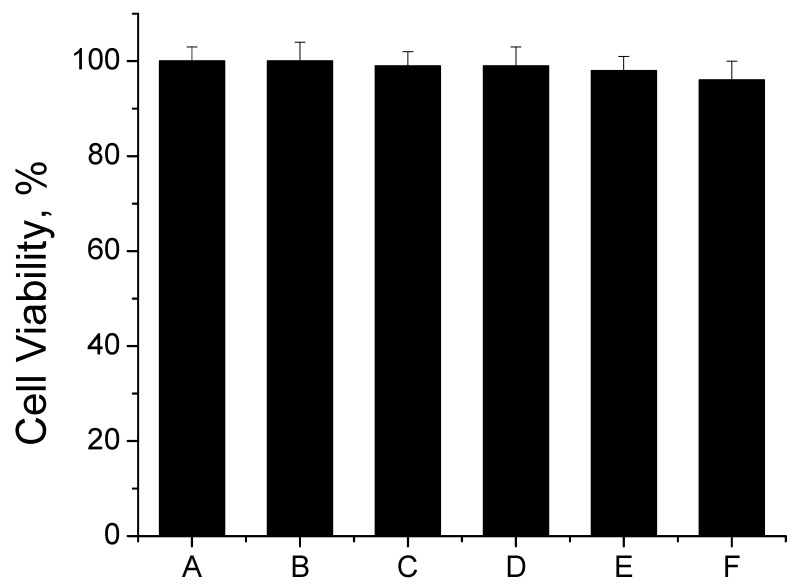
Cell viability of (A) TCPS, (B) CPMA, (C) CPMAZnO-1, (D) CPMAZnO-2, (E) CPMAZnO-3, and (F) CPMAZnO-4 films.

**Figure 10 jfb-14-00228-f010:**
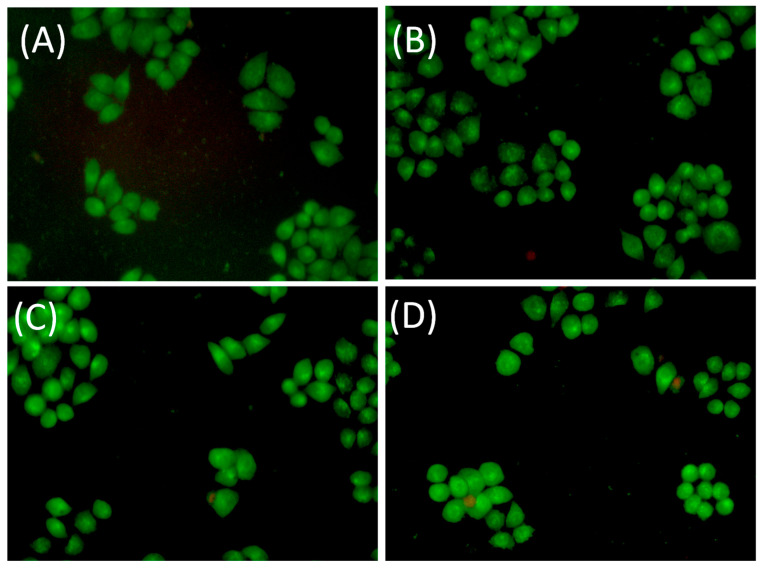
Images of AO–EB stained C3H10T1/2 cells treated with (**A**) CPMAZnO-1, (**B**) CPMAZnO-2, (**C**) CPMAZnO-3, and (**D**) CPMAZnO-4 films.

**Table 1 jfb-14-00228-t001:** Types of CPMA ZnO films and precursor concentrations.

Type of PECs	Chitosan	PMA	Zn(CH_3_COO)_2_·2H_2_O
	(Wt. %)	(Wt. %)	(mmol)
CPMA	2	2	0
CPMAZnO-1	2	2	50
CPMAZnO-2	2	2	75
CPMAZnO-3	2	2	100
CPMAZnO-4	2	2	150

## Data Availability

Not applicable.

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
