# Peer review of "Zinc Oxide–Incorporated Chitosan–Poly(methacrylic Acid) Polyelectrolyte Complex as a Wound Healing Material"

_jfb, 2023, doi:10.3390/jfb14040228_

Round 1

Reviewer 1 Report

The current manuscript explains about materials for wound healing. The idea is good but still it needs some corrections.

1. Abstract should be revised and include the main findings of the study.

2. why chitosan with MW 250 KDa is selected?? Explains this point in introduction section.

3. Antibacterial study was performed against two strains. is there any relation of these strains with wound healing?? if yes then explains in discussion part.

4. is there any reference stating that porosity about 64% is  more suitable for effective wound healing?? explain is porosity section.

5. Results of antibacterial study if presented in graphical/table form is more easy to understand.

Regards

Author Response

Answers for Reviewers’ comments:

Reviewer -1:

Comment -1. Abstract should be revised and include the main findings of the study.

Answer: As suggested, the abstract is revised with the main findings in the revised manuscript (page 1, line 14-18).

Comment -2. why chitosan with MW 250 KDa is selected?? Explains this point in introduction section.

Answer: Since the MW is high, the polymer will exhibit an improved mechanical strength and PEC with anionic polymers. Hence, we utilized chitosan with MW of 250 KDa in this study. Since MW is not a subject of our interest, this is not discussed in the introduction.

Comment -3. Antibacterial study was performed against two strains. is there any relation of these strains with wound healing?? if yes then explains in discussion part.

Answer: Wound infection on the skin surface is most often caused by bacteria such as Staphylococcus aureus (S. aureus) and Micrococcus sp. Hence, in this study, the antibacterial property of CPMAZnO films against S. aureus and Micrococcus sp was determined. Required information is given in the revised manuscript (page 6, line 154-156).

Comment -4. is there any reference stating that porosity about 64% is  more suitable for effective wound healing?? explain is porosity section.

Answer: Reference and explanation are included in the revised manuscript (page 13, line 287-289; page 21, reference 27).

Comment -5. Results of antibacterial study if presented in graphical/table form is more easy to understand.

Answer: Only CPMAZnO-4 has shown the zone of inhibition. This result is given in the text (page 16, line 338). Hence, result is not presented in graphical/table form.  

Reviewer 2 Report

1. Please avoid using abbreviations in abstract.

2. Also write complete title in heading and subheading

3. In swelling study, author written at fixed time period, please write the time frame.

4. Kindly cite the biodegradation process.

5. Why the specific bacteria were selected? suggested to add relevance on selection of bugs.

6. Suggested how the biocompatibility of material was tested, in this case citation does not make sense exactly. Chitosan practically won't degrade in cell culture media, so how it was tested.  

7.  Although the treatment for test and control was same for cell seeding, the density is very high for 96-well considering incubation duration 

8. 10 mL of media containing with 100 mf of film, is it cell culture media or sterile Phosphate buffer

10. During curing process? what author wants to tell

11. Suggested to calculated the size of nano structure using Debye Scherrer  equations for the obtained values of XRD spectra

12. Suggested to write discussion why pore size varied, usually controlled freeze-drying cause change in pore structure 

13. Suggested to mark the zone of inhibition properly, so reader can understand properly.

14. What was effect on the cell viability on increment of ZNO and why it happen please discuss with citations.

15. Overall discussion needs improvement 

There are some correction suggested in attached file, please refer it too.

Author Response

Answers for Reviewers’ comments:

Reviewer -2:

Comments and Suggestions for Authors

Comment -1. Please avoid using abbreviations in abstract.

Answer: As suggested, abbreviations were removed (Page 1, lines 10 - 13, 17, 18).

Comment -2. Also write complete title in heading and subheading

Answer: As suggested, complete title is given in the revised manuscript (page 1, line 1).

Comment -3. In swelling study, author written at fixed time period, please write the time frame.

Answer: Time frame is given in the revised manuscript (Page 13, line 291, 292)

Comment -4. Kindly cite the biodegradation process.

Answer: As suggested, cited the biodegradation process in the revised manuscript (page 5, line 149, ref. [13]).

Comment -5. Why the specific bacteria were selected? suggested to add relevance on selection of bugs.

Answer: Wound infection on the skin surface is most often caused by bacteria such as Staphylococcus aureus (S. aureus) and Micrococcus sp. Hence, in this study, the antibacterial property of CPMAZnO films against S. aureus and Micrococcus sp was determined. Required information is given in the revised manuscript (page 6, line 154-156).

Comment -6. Suggested how the biocompatibility of material was tested, in this case citation does not make sense exactly. Chitosan practically won't degrade in cell culture media, so how it was tested.  

Answer: Biocompatibility study was conducted by indirect MTT assay. Complete procedure is given in the revised manuscript as suggested (page 6, line 164-173).

Comment -7.  Although the treatment for test and control was same for cell seeding, the density is very high for 96-well considering incubation duration.

Answer: The procedure is followed based on the reference [15].

Comment -8. 10 mL of media containing with 100 mf of film, is it cell culture media or sterile Phosphate buffer.

Answer: That is phosphate buffer. Clear information is given in the revised manuscript (page 6, line 178-179).

Comment -9. During curing process? what author wants to tell

Answer: To avoid confusion, sentence has been revised in the revised manuscript (page 4, line 110).

Comment -10. Suggested to calculate the size of nano structure using Debye Scherrer  equations for the obtained values of XRD spectra.

Answer: As suggested, crystallite size of ZnO was calculated using Debye Scherrer  equations and the result is given in the revised manuscript (page 10, line 238-240).

Comment -11. Suggested to write discussion why pore size varied, usually controlled freeze-drying cause change in pore structure.

Answer:  Variation in the pore size may be due to the presence of ZnO. As suggested, the relevant discussion is given in the revised manuscript (page 12, line 269-273).

Comment -12. Suggested to mark the zone of inhibition properly, so reader can understand properly.

Answer: As suggested, the inhibition zone is marked in the Figure 8 (page 16).

Comment -13. What was effect on the cell viability on increment of ZNO and why it happen please discuss with citations.

Answer: The effect of ZnO on cell viability is given and discussed in the revised manuscript (page 16, 17, line 351-354).

Comment -14. Overall discussion needs improvement.

Answer: As suggested, discussion sections were improved in the revised manuscript .

Comment -15. There are some corrections suggested in attached file, please refer it too.

Answer: As suggested, all the marked corrections were rectified in the revised manuscript.

Reviewer 3 Report

I have reviewed manuscript entitled "ZnO-incorporated Chitosan-Poly(methacrylic acid) Polyelectrolyte Complex as a Wound Healing Material". Work is good and have sound scientific value. In order to improve this work, i would like that authors must address following queries / suggestions;

1) In preparation of polymeric films kindly elaborate conditions of freeze drying and recheck temperature of 120oC? also mention the dimensions of petri dishes used.

2) Figure 1. Change superscript zero with degree signs and also everywhere.

3) Since CPMA films were prepared as a result of freeze drying that usually leaves porous products but in section 3.2 you are saying there were no porous surfaces in case of CPMA films? kindly address this

4) Mention the pH of swelling media?

5) Add error bars in Figure 6.

6) Figure 8 is not showing any zone of inhibition practically. 

7) While dealing with polymeric films, kindly add results of mechanical strength, thickness of films etc.

8) Section 2.3 add details of fixed time intervals and recheck swelling degree equation?

9) Update reference list by the references from last five years only. 

Author Response

Answers for Reviewers’ comments:

Reviewer 3:

Comments and Suggestions for Authors

I have reviewed manuscript entitled "ZnO-incorporated Chitosan-Poly(methacrylic acid) Polyelectrolyte Complex as a Wound Healing Material". Work is good and have sound scientific value. In order to improve this work, i would like that authors must address following queries / suggestions;

Comment -1. In preparation of polymeric films kindly elaborate conditions of freeze drying and recheck temperature of 120oC? also mention the dimensions of petri dishes used.

Answer: The condition of freeze drying and petri dish size are given in the revised manuscript. The temperature 120°C is used for drying process (page 4, line 109 -110).

Comment -2. Figure 1. Change superscript zero with degree signs and also everywhere.

Answer: As suggested, superscript zero was replaced with degree signs everywhere.

Comment -3. Since CPMA films were prepared as a result of freeze drying that usually leaves porous products but in section 3.2 you are saying there were no porous surfaces in case of CPMA films? kindly address this.

Answer: The CPMA films presented rough morphology without any pores on its surface. This observation can be due to the raise in polymer concentration, which determined a major challenge to pore growth on the sample surface. The required information is given in the revised manuscript (page 12, line 264-265).

Comment -4. Mention the pH of swelling media?

Answer: As suggested, the pH of swelling media is given in the revised manuscript (page 5, line 136).

Comment -5. Add error bars in Figure 6.

Answer: As suggested, error bars are given in Figure 6 (page 14).

Comment -6. Figure 8 is not showing any zone of inhibition practically. 

Answer: Only CPMAZnO-4 has shown the zone of inhibition, which is marked in Figure 8 (page 16).  

Comment -7. While dealing with polymeric films, kindly add results of mechanical strength, thickness of films etc.

Answer: As suggested, tensile strength and elongation at break were given in the revised manuscript (page 14, line 312-317).

Comment -8. Section 2.3 add details of fixed time intervals and recheck swelling degree equation?

Answer: As suggested, details of fixed time intervals were given and swelling degree equation was rechecked and corrected in the revised manuscript (page 5, line 137, 143).

Comment -9. Update reference list by the references from last five years only.

Answer: As suggested, references from last five years were updated in the revised manuscript (page 19-22, References 4, 8, 9, 12, 13, 16, and 28)

Reviewer 4 Report

1.      The authors present a new ZnO-incorporated chitosan-poly (methacrylic acid) polyelectrolyte complex. They guide the reader with enough measurements to the conclusion that this material has been produced.

2.      The article in ending with some advantages that the new material could bring in the future

3.      The presentation and evaluation of some methods are weak and not sufficiently compared with the current literature

4.      Synthesis of samples can be given as a flowchart/schematic diagram

5.      The author must be provided with a visible scale bar for SEM images

6.      The quality of the figure should be improved

7.      The discussion section should be expanded to highlight the scientific contribution of this study to this field.

8.      The accuracy of the measurements of the technology should be presented

9.      What are possible technology-oriented applications of the work for commercialization purposes?

10.  In conclusion, instead of only fact lists, more descriptions including future perspectives had better be added.

11.  English of the manuscript needs a significant revision.

Author Response

Answers for Reviewers’ comments:

Reviewer 4:

Comments and Suggestions for Authors

Comment -1.      The authors present a new ZnO-incorporated chitosan-poly (methacrylic acid) polyelectrolyte complex. They guide the reader with enough measurements to the conclusion that this material has been produced.

Answer: As suggested, the conclusion part is revised (page 19, line 386-392).

Comment -2.      The article in ending with some advantages that the new material could bring in the future.

Answer: The future direction of the work is given in the revised manuscript (page 19, line 386-392).

Comment -3.      The presentation and evaluation of some methods are weak and not sufficiently compared with the current literature.

Answer:  The presentation and evaluation methods have been improved in the revised manuscript (page 6, line 164-73; page 12, line 264-266, 269-272).

Comment -4.      Synthesis of samples can be given as a flowchart/schematic diagram.

Answer: As suggested, synthesis of samples is given as a schematic diagram (page 7, figure 1).

Comment -5.      The author must be provided with a visible scale bar for SEM images.

Answer: As suggested, visible scale bar is given for SEM images in the revised manuscript (page 12, Figure 5).

Comment -6.      The quality of the figure should be improved.

Answer: As suggested, quality of the figures improved in the revised manuscript (page 7, figure 1; page 14, figure 6; page 16, figure 8).

Comment -7.      The discussion section should be expanded to highlight the scientific contribution of this study to this field.

Answer: As suggested, the discussion section was expanded in the revised manuscript (page 14, line 312-319; page 16, line 349-354).

Comment -8.      The accuracy of the measurements of the technology should be presented.

Answer: As suggested, the accuracy of the measurements was given in the revised manuscript (page 13, line 284-286; page 14, line 317).

Comment -9.      What are possible technology-oriented applications of the work for commercialization purposes?

Answer: The developed materials could be useful for wound healing applications. Since the raw material cost is low and processing is easy, the large-scale production of wound dressings based on the proposed martials is possible in near future.

Comment -10.  In conclusion, instead of only fact lists, more descriptions including future perspectives had better be added.

Answer: As suggested, more descriptions for future perspectives is given in the revised manuscript (page 19, line 386-392).

Comment -11.  English of the manuscript needs a significant revision.

Answer: As suggested, the language of manuscript was completely revised. 

Round 2

Reviewer 1 Report

The manuscript is revised accordingly, and is now suitable for publication.

Regards

Reviewer 2 Report

The authors have reflected all the suggestions, I recommend to editor that the manuscript can be accepted in present form. 

Reviewer 4 Report

The authors addressed each comment in detail and made appropriate changes. I recommend its publication in the present form.